# The Need for Culturally Appropriate Food Allergy Management Strategies: The Indian Milk Ladder

**DOI:** 10.3390/nu15183921

**Published:** 2023-09-09

**Authors:** Somashekara Hosaagrahara Ramakrishna, Neil Shah, Bhaswati C. Acharyya, Emmany Durairaj, Lalit Verma, Srinivas Sankaranarayanan, Nishant Wadhwa, Carina Venter

**Affiliations:** 1Department of Paediatric Hepatology & Gastroenterology, Gleneagles Global Health City, Chennai 600100, India; somu_cmch@yahoo.com; 2The Portland Hospital, London W1W 5AH, UK; neil.shah@rb.com; 3Reckitt Nutrition, Slough SL1 3UH, UK; 4Department of Paediatric Hepatology & Gastroenterology, AMRI Hospitals, Kolkata 700028, India; bachharya21@gmail.com; 5Department of Paediatric Hepatology & Gastroenterology, Institute of Child Health, Kolkata 700017, India; 6Department of Clinical Nutrition, Mazumdar Shaw Medical Centre, Narayana Health City, Bengaluru 560099, India; dremmans@gmail.com; 7Department of Paediatric Hepatology & Gastroenterology, Global Hospitals, Mumbai 400012, India; drvermalalit@gmail.com; 8Department of Paediatric Hepatology & Gastroenterology, Kanchi Kamakoti CHILDS Trust Hospital, Chennai 600024, India; dr.srinis@gmail.com; 9Department of Paediatric Hepatology & Gastroenterology, Apollo Children Hospital, Chennai 600006, India; 10Institute of Child Health, Division of Paediatric Gastroenterology, Hepatology & Nutrition, Sir Gangaram Hospital, New Delhi 110060, India; docnish@gmail.com; 11Paediatrics, Section of Allergy & Immunology, Denver School of Medicine, University of Colorado, Denver, CO 80204, USA; 12Paediatrics, Section of Allergy & Immunology, Children’s Hospital Colorado, Denver, CO 80218, USA

**Keywords:** cow’s milk allergy, CMA, milk ladder, tolerance acquisition, baked food

## Abstract

Background: Cow’s milk allergy (CMA) is one of the most common and complex food allergies affecting children worldwide and, with a few exceptions, presents in the first few months of life. Baked-milk-containing diets are well tolerated in the majority of milk-allergic children and allow dietary restrictions to be relaxed. In addition, the early introduction of tolerated forms of allergenic foods to an infant’s diet in small amounts may enhance the outgrowth of their milk allergy through oral tolerance induction. The methods of milk introduction vary widely across the globe. Methods: We convened an expert group to develop a comprehensive milk ladder based on the calculated milk protein content of Indian foods. To validate the milk ladder, the foods chosen for the ladder were analyzed and the ladder was re-evaluated based on the cooked milk protein content. Results: Combining expert consensus and validation of milk protein content, we created the world’s first milk ladder containing Indian foods. This is the first ladder that provides information on the timing and temperature of cooking, with validated milk protein content. Conclusions: This is the first milk ladder based on the unique features of Indian food habits built by the consensus of Indian experts along with international collaboration with laboratory quantification of milk protein in each step. We believe the “The Indian Milk Ladder” will be a very helpful tool for pediatricians helping manage CMA in children as well as their parents and caregivers, not only in India, but in countries world-wide where these foods are commonly consumed.

## 1. Introduction

Cow’s milk allergy (CMA) is the most common and complex presentation of food allergy in early childhood [1,2,3,4]. Cow’s milk allergy can be divided into IgE- and non-IgE-mediated forms of food allergy. CMA has an estimated prevalence in developed countries ranging from 0.5% to 3% at the age of 1 year [5]. Non-IgE-mediated cow’s milk allergy has a prevalence of less than 1% in children [6]. There has been no epidemiologic research on the prevalence of food allergies, including CMA, among Indian children. A hospital-based study from India reported CMA as a cause of malabsorption syndrome in 6% of children of all ages and 13% of children below 2 years with chronic diarrhea [7]. In another study from India, CMA was the cause of chronic diarrhea in 35% of children below 3 years of age [8]. Mandal J et al. [9] reported milk hypersensitivity in 25% of Indian children aged 5 to 15 years who presented with food allergies.

Non-IgE-mediated CMA is again classified into the food protein-induced enterocolitis syndrome (FPIES), eosinophilic gastro-intestinal disorders, food protein-induced proctocolitis [10] and mild to moderate non-IgE-mediated food allergies [4,11,12]. IgE-mediated CMA is diagnosed based on a combination of clinical history, specific IgE tests/skin prick tests and oral food challenges when required [10]. CMA-related FPIESs are diagnosed by a clinical history and/or oral food challenge [13]. Eosinophilic gastro-intestinal disorders are diagnosed by performing endoscopies followed by cow’s milk avoidance, reintroductions and follow-up endoscopies [14]. Food protein-induced proctocolitis and cow’s milk-induced enteropathies are mainly diagnosed based on clinical features and the patient’s medical history and cow’s milk avoidance [15].

Mild to moderate non-IgE-mediated CMAs were described for the first time in 2013 by a group of clinicians from the United Kingdom [12]. The authors suggested an algorithm for the diagnosis and management of mild to moderate non-IgE-mediated CMA. Infants with mild to moderate non-IgE-mediated CMA present with a combination of skin, gastrointestinal and sometimes respiratory symptoms in the absence of growth faltering and severe symptoms. The diagnosis is based on a clinical history, removal of cow’s milk protein and rechallenge of cow’s milk upon symptom resolution (usually 2–4 weeks after removal of cow’s milk protein). After a period of avoidance, cow’s milk can be reintroduced at home in a sequential manner starting from the lowest dose and allergenicity to the highest dose and allergenicity over a time period, which can be individualized for each case.

The original ladder for use in children with mild to moderate non-IgE-mediated CMA was developed by a group of UK health care professionals calculating the amount and cooking temperature of 12 foods commonly consumed in the UK (malted milk biscuits, digestive biscuits, cupcakes/mini muffins, scotch pancakes, shepherd’s pie, lasagna, pizza, milk chocolate, yogurt, cheese, sterilized and pasteurized milk) [12]. It soon became apparent that the ladder was used world-wide and an amended version of the ladder, using more internationally accepted foods and only six steps, was developed (biscuits/cookies, muffins, pancakes, cheese, yogurt, and pasteurized milk) [11]. Yet, it was clear that the foods in the ladder did not represent food preferences of all cultures.

Subsequently, many milk ladders were developed based on local dietary patterns and are in clinical use. Recently, a Mediterranean milk ladder based on the principles of the Mediterranean eating pattern was developed. The protein content of cooked protein in the final food product in each step of the ladder matched with that of the IMAP ladder. Different recipes for the various steps were provided to increase acceptability and variety. Additionally, the total milk protein content, casein content, and beta-lactoglobulin content of cooked food were estimated by enzyme-linked immunosorbent assay (ELISA) [16]. Many modified forms of the milk ladder are widely used internationally and, in some cases, include recipes with local preparation differences [17,18,19]. Chomyn, A et al. [20] independently proposed a Canadian version of cow’s milk and egg food ladders as they felt that the European versions of food ladders had limited applicability to the Canadian diet, as they did not include foods that are commonly consumed in many Canadian households.

Data from the UK have shown that mothers were confused about the steps in the milk ladder and concerned about the limited number of foods that might not necessarily reflect the child’s food preferences, which led to the adaptation of the original ladder [21]. A recent rostrum article highlights the fact that milk ladders are increasingly being used in the care of children with IgE-mediated food allergy [12]. This rostrum further emphasizes the importance of milk ladders being simple with a limited number of foods per step, culturally and age appropriate, preferably healthy, along with attached recipes including the time and temperature of heating and also the milk protein content to be specified. The levels of protein present in the final product should be tested [22].

Indian food is different from the rest of the world, not only in taste, but also in cooking methods and ingredients used. It reflects a perfect blend of various cultures and ages. Kashmiri dishes reflect strong central Asian influences with rice as the staple diet compared to wheat as the staple diet in the rest of Northern India. Western Indian foods contain a range of lentil dishes and pickles or preserves. Food intake in Eastern India is dominated by fish and rice dishes. Southern Indian food also often contains fish as an ingredient as well as lamb, prawns and coconut. However, central to all Indian cooking is sweet, milky desserts. We have therefore considered the sugar and fat content of foods, but included some foods higher in fat and sugar content to adhere to the Indian palate. The study aimed to (1) develop a milk ladder based on food cultures in India, (2) consider and be mindful of the sugar and fat content in the Indian foods included in the ladder and (3) calculate and test the amount of cooked milk protein in the different Indian foods included in the ladder.

## 2. Materials and Methods

### 2.1. Formation of Expert Group

Taking into consideration the prior knowledge described previously, an expert group experienced in managing CMA was convened to develop a milk ladder using Indian foods. The group was composed of a dietitian and Indian pediatric gastroenterologists (from all four regions of India—North, South, East and West) as well as an international pediatric gastroenterologist and pediatric allergy specialist dietician.

### 2.2. Initial Development of the Indian Milk Ladder

The expert group members from the different regions in India were asked to provide a list of the foods their patients commonly want to introduce when performing introduction via a ladder approach. The group had a number of teleconferences where the foods were discussed and overlap in food choices were identified in order to develop the final list of foods in the ladder. A list of common foods reflecting the different cultures in India was drawn up, analyzed for their milk protein content, temperature and timing of heating, sugar and fat content (Table 1). The first list of foods included Maida Diamond Biscuits, Gulab Jamun, Ragi/Poha Dosa, Halwa, Rava Idli, rice/sago/semolina pudding, paneer, scrambled egg with milk and Rava Laddoo. After discussion of the fat, sugar and allergen content, the working (initial) Indian Milk Ladder was designed by consensus (Table 1). Foods that are prevalent across different geographical locations and cultures were included.

### 2.3. Testing of the Milk Protein Content of Food Included in the Milk Ladder

To rationalize the steps of the Indian Milk Ladder, we first calculated and then tested the content of milk protein in the different Indian foods included in the ladder. Food items were cooked freshly as per the recipe (detailed in Appendix A), which specified the ingredients, method of cooking and amount of milk/milk product to be added.

Kjeldahl nitrogen analysis, Association of Official Analytical Chemists (AOAC) methods [23] and ELISA [24] were used to determine the cooked milk protein content. The details of the ELISA kits are as follows: RIDASCREEN^®^FAST Milk (Art No R4652), R-Biopharm AG, Darmstadt, Germany, Lot Number: 25470 Expiry: 11/2021.

Total nitrogen analysis by the Kjeldahl method, the international reference method for determining protein in milk and milk-based products due to its high precision and good reproducibility, which indirectly measures protein content by determining the total nitrogen content of the milk product, was carried out. The total nitrogen was converted into the percentage of protein and accounts for the nitrogen content of the average amino acid composition present in the milk [23,25]. For the milk-derived products tested, including paneer, cheese, yogurt, Rasagulla, and Shrikhand, the protein content directly represented the total milk protein and the AOAC 2001.11 [26] method was used. For products containing milk along with other ingredients (Gajar Halwa, Gulab Jamun, Ragi Sari, Rice Kheer), due to the ingredient variability and presence of protein sources other than milk, the milk proteins were determined as casein and albumin content per the AOAC 939.02 [27] method.

The AOAC method is not recommended to determine the milk protein content of samples that are presumed to contain less than 0.1% of milk proteins. The sandwich enzyme-linked immunoassay (ELISA) for the quantitative determination of milk protein in food was used to determine milk protein [24] for the following items: biscuit, Maida Biscuit, cookie, pan cake, Ragi Dosa, Rava Idli and muffin. The ELISA method uses antibodies against caseins and beta-lactoglobulins, and thus it is a direct measure of milk protein content in the samples.

Milk protein was estimated for three different samples at three different times by the methods described above. The average of three values was taken as the final result. Mean and standard deviations were calculated.

## 3. Results

### 3.1. Expert Consensus

The initial milk ladder was designed by consensus by an expert committee by calculating the amount of milk in food and accounting for the cooking temperature and duration (Table 1). The principle used to design each step of the ladder is outlined in the list below:

Step 1 and 2: Gradually increasing the amount of protein and cooking time;

Step 3: Slightly lower amount of protein and heating;

Step 4: Increased protein with increased cooking time;

Step 5: No cooking;

Step 6: Pasteurized milk.

### 3.2. Validation of Milk Protein in Selected Foods

Following expert consensus, the cooked milk protein content of foods included in the ladder was analyzed in the laboratory as described. The cooked milk protein contents of the analyzed foods are outlined in Table 2.

### 3.3. Adaptation of Sugar Content

The foods identified by the expert group were typically high in fat and sugar. SHR and ED spent considerable time to reduce the sugar and fat content of foods to be more acceptable according to nutrition standards but still palatable. The DG USA recommends less than 10% kcal from sugar for children over 2 years, which is 25 g sugar per day. We have reduced the sugar content as indicated in Table 3.

### 3.4. Consideration of Food Matrix

Bloom et al. [28] showed that a wheat matrix decreases IgE antibody binding to milk proteins. In addition, the chemical reactions between proteins and fat and sugars in recipes may also reduce allergenicity [29]. Bacteria used in yogurt production may also affect allergenicity but this has not been well studied. Our awareness of all these factors led to not only calculating the milk content of the ladder but also scientifically quantifying the milk content of the ladder foods within their complex matrices.

### 3.5. Finalization of the Indian Milk Ladder

The results obtained by the laboratory milk protein estimations were compared with that of the original milk ladder designed by consensus. The final milk ladder was prepared by rearranging the food products in the ladder according to the amount of protein present in the food item as determined by laboratory testing. The ladder contains six steps (Table 4 and Figure 1). Alternative Indian food to reflect cultural differences is provided in each step. Detailed recipes of foods in the ladder and method of reintroduction will be provided to parents/caretakers (Appendix A: Detailed Recipes; Appendix A: Practical guide to use the milk ladder).

## 4. Discussion

The gradual reintroduction of baked milk products and milk in children with non-IgE-mediated CMA carried out at home is called the “Milk Ladder”. The milk ladder is based on the knowledge that the addition of baked milk to the diet of children that can tolerate such foods appears to accelerate the development of milk tolerance compared with strict avoidance [17,18]. We developed the first milk ladder based on the unique features of Indian food habits through the consensus of Indian experts along with international collaboration. This is the world’s first milk ladder that quantifies the amount of milk protein per step.

Heat treatment has long been recognized to have the potential to alter the allergenicity of proteins. Now, it is well recognized that the processing of food proteins alters their structure and potential to induce allergy [28,30]. The allergenic characteristics of a protein are determined by the epitopes formed by the sequential amino acids and epitopes arising from the three-dimensional shape of the protein called conformational epitopes. Heating causes a loss of the conformational epitopes and largely preserves the sequential epitopes [31].

It is generally accepted that infants with a proven diagnosis of cow’s milk allergy should remain on a cow’s-milk-protein-free diet until 9–12 months of age and for at least 6 months [32] prior to the re-introduction of cow’s milk into their diets. However, up to 90% of children with CMA may tolerate baked-milk-containing foods such as muffins and cupcakes [33]. The inclusion of baked milk products in children’s diets is suggested to accelerate the development of tolerance to unheated milk compared to a strict milk avoidance approach. A significant increase in IgG4 against milk casein in children who are tolerant to baked milk has been reported, similar to those treated with milk oral immunotherapy [34,35]. Moreover, the consumption of baked milk is suggested to enhance the quality of life of these children by removing unnecessary dietary restrictions and to change the natural history of CMA by promoting the tolerance acquisition to regular cow’s milk [17,36].

However, we agree that the evidence for the introduction of baked milk products in an infant’s diet to accelerate the development of tolerance is of low quality, as highlighted in a systematic review by Labert R et al. [37] They cited the lack of randomized, controlled trial data and a comparator cohort within the study groups of various publications to determine whether or not baked allergen speeds up the resolution of allergy. They found that studies were mainly observational and merely gave the information that if the child is already tolerant to baked milk, they are more likely to have outgrown their food allergy at any particular time-point than a child who is not tolerant to baked cow’s milk.

Another factor that may hinder clinicians from using a milk ladder for the reintroduction of baked milk products is the occurrence of anaphylaxis. However, most studies of baked milk challenge in children did not report any anaphylactic reactions [38,39], except for one study that report mild to moderate anaphylactic reactions in 4.7% of children [17]. 

All milk proteins have the potential to act as allergens. In the current study, we designed the first milk ladder in the Indian subcontinent considering the cultural diversity in India, the amount of milk protein in the food item, the time/duration and temperature of heating, as well as the effect of other ingredients on the milk protein. The cooked milk protein in each chosen food substance was analyzed by standard laboratory methods. We estimated only the total milk protein content of foods as we know that both casein and whey are equally implicated in the pathogenesis of milk allergy [40,41]. In children, major milk allergens are suggested to be caseins, β-lactoglobulin and α-lactalbumin [42]. The calculated protein content of foods in step 1, step 2, step 3 and step 4 did not correlate well with the laboratory estimation of milk protein. However, the foods in step 5 (milk-derived products) of the Indian Milk Ladder correlated well with the laboratory estimation. Based on the laboratory results, we rearranged the foods of the milk ladder in step 2, step 3 and step 4. As the foods in step 2, 3 and 4 contained other ingredients along with significant amounts of milk, the interference of the matrix would have contributed to this variation. Previous studies have shown that incorporating other substances (a matrix) along with milk while cooking might promote the formation of complex milk-food components [43]. We hypothesize these milk-food components may lead to interference in the laboratory estimation of milk protein, leading to a difference between the calculated and estimated milk protein content in cooked foods. It is also known that milk food components induce a modulation of the immunoreactivity towards milk allergens [43]. Based on the results of the milk protein content of different food substances, the ladder was re-designed by arranging the food items in increasing order of protein content.

It is generally accepted that the ladder approach can be used safely in children with mild to moderate non-IgE-mediated cow’s milk allergy such as proctocolitis. It is however also known that the milk ladder is being used in children with IgE-mediated cow’s milk allergy outside of the clinical setting [21]. A recent article [23] suggests that the milk ladder can be used safely in individuals with (1) non-IgE-mediated allergy (excluding FPIES), (2) IgE-mediated food allergy with prior mild, non-anaphylactic reactions, (3) no diagnosis of asthma individuals, but considered for individuals with stable, treated asthma, (4) the ability to understand and comply with the instructions provided, (5) a high previous reaction threshold, (6) low or decreasing skin prick test wheal or serum specific-IgE levels; (7) younger patients or individuals of any age with limited co-existing allergies. However, because we are employing IML for home-based re-introduction, we strongly advise that it is used only for children with non-IgE-mediated milk allergies.

One limitation of the study is that we only measured the total milk protein content and not the whey and casein fractions. Future analysis should include measuring the whey and casein fractions of baked and processed milk products.

## 5. Conclusions

This is the first attempt to design a milk ladder taking into account cultural food preferences such as the foods consumed in India and elsewhere. We have developed a milk ladder with clear instructions for use in children who are allergic to cow’s milk. This is also the first ladder that provides information on the timing and temperature of cooking, and validated milk protein content.

We hope that the “The Indian Milk Ladder” will be a very valuable resource and helpful tool for the pediatricians caring for children with CMA and also for parents of these children worldwide, particularly clinicians who are not very well informed regarding Indian foods.

## Figures and Tables

**Figure 1 nutrients-15-03921-f001:**
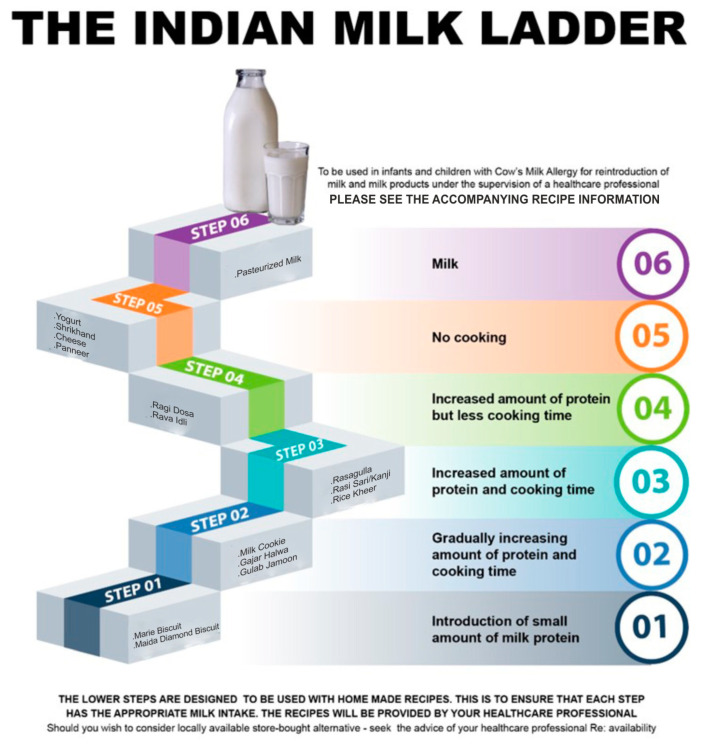
The Indian Milk Ladder.

**Table 1 nutrients-15-03921-t001:** Milk ladder built by consensus.

Step	Food	Milk/Serving	Milk Protein/ Serving	Cooking Temperature (°C)	Cooking Time
Step 1 and 2: Gradually increasing amount of protein with increasing cooking time					
Step 1	Marie Biscuit		0.050 g	180	15 min
	Maida Diamond Biscuits	1.25 mL	0.040 g	180	15 min
Step 2	Milk Cookie	1.8 g milk powder	0.37 g	180	10 min
	Halwa	25 mL	0.75 g	Boiled	25 min
	Ragi Sari/Kanji (with milk)	25 mL	0.75 g	Boiled	30 min
Step 3: Slightly lower amount of protein and lower heating time					
Step 3	Ragi Dosa	21 mL	0.74 g	Fry	2–4 min
	Rava Idli	20 mL	0.70 g	Steam	15 min
Step 4: Increased Protein and Increased Cooking time too					
Step 4	Gulab Jamun (dry)	8.3 g milk powder	2.15 g	Boiled + Fried	20 min
	Rasagullah	100 mL	3.5 g	Boiled	30 min
	Rice Kheer	100 mL	3.5 g	Boiled	30 min
Step 5: No Cooking					
Step 5	Paneer		3.7 g	No cooking But through cheese-making process	
	Shrikand or Yoghurt	180 mL	6.0	No cooking	
Step 6: Pasteurized Milk					
Step 6	Milk	¼ cup	2.17 g		
		½ cup	4.34 g		
		1 cup	8.68 g		

Legend: Initial milk ladder built by consensus. Milk protein content was calculated manually by going through each recipe and estimating the quantity of milk present per serving. g = grams; °C = Celsius; mL = milliliters; min = minutes.

**Table 2 nutrients-15-03921-t002:** Cooked milk protein content of foods in the Indian Milk Ladder (in grams/100 g of food).

Sample Name	Method	Reading 1	Reading 2	Reading 3	Mean	SD
Biscuit	ELISA	0.005	0.005	0.005	0.005	0
Maida Biscuits	ELISA	0.006	0.005	0.006	0.005	0.0005
Shrikhand	AOAC 939.02	0.28	0.28	0.28	0.28	0
Cookie	ELISA	0.32	0.38	0.38	0.36	0.0346
Pan Cake	ELISA	0.34	0.35	0.37	0.353	0.0152
Gajar Halwa	AOAC 939.02	0.41	0.41	0.41	0.41	0
GulabJamun	AOAC 939.02	1.23	1.23	1.23	1.23	0
Rasagulla	AOAC 2001.11	2.49	2.67	2.49	2.55	0.1039
Ragi Sari	AOAC 939.02	2.73	2.59	2.73	2.68	0.0808
Rice Kheer	AOAC 939.02	2.87	2.87	2.73	2.82	0.0808
Ragi Dosa	ELISA	2.92	2.997	2.87	2.92	0.0639
Rava Idli	ELISA	3.12	3.08	3.2	3.13	0.0611
Muffin	ELISA	4.63	4.08	4.36	4.35	0.2750
Yogurt	AOAC 2001.11	7.33	7.21	7.16	7.23	0.0873
Cheese	AOAC 2001.11	21.15	21.28	21.1	21.17	0.0929
Paneer	AOAC 2001.11	21.37	21.49	21.49	21.45	0.0692

Legend: The foods from the ladder built by consensus were analyzed for cooked milk protein content in the laboratory. Three different samples were analyzed at three different times. AOAC = Association of Official Analytical Chemists; SD = standard deviation.

**Table 3 nutrients-15-03921-t003:** Adapted sugar content of foods high in sugar.

Milk Ladder Foods	Sugar per Portion in Original Recipes	Sugar Content in Adapted Recipes	Reduction in Sugar per Portion
Maida Diamond Biscuit	1.5 g	0.75 g	0.75 g
Milk Cookie	11.25 g	1.9 g	9.6 g
Gajar Halwa	25 g	6 g	19 g
Gulab Jamun	20 g	5 g	15 g
Rasagulla	33 g	13 g	20 g
Ragi Sair/Kanji	5 g	5 g	Nil
Rice Kheer	31.25 g	12.5 g	18.75 g
Srikhand	30 g	7.5 g	22.5 g

Legend: The recipes of foods containing high sugar were adapted by reducing the amount of sugar.

**Table 4 nutrients-15-03921-t004:** The final Indian Milk Ladder.

Step	Food	Cooked Milk Protein Content (g/100 g)	Recommended Portion per Serving	Cooking Temperature (°C)	Cooking Time
Step 1: Introduction of small amount of milk protein					
Step 1	Marie Biscuit	0.005	½ Biscuit to start with & build up gradually	Commercial Preparation	
	Maida Dimond Biscuit	0.006		180	15 min
Step 2: Gradually increasing amount of protein and cooking time					
Step 2	Milk Cookie	0.35	Start with ¼ portion & increase gradually	180	12 min
	Gajar Halwa	0.410		Boil	25 min
	Gulab Jamun (dry)	1.230		Boil + Fry	20 + 5 min
Step 3: Increasing amount of protein and cooking time					
Step 3	Rasgulla	2.550	Start with ¼ portion increase gradually	Boil	30 min
	Ragi Sari/Kanji	2.686		Boil	30 min
	Rice Kheer	2.823		Boil	30 min
Step 4: Increasing amount of protein but less cooking time					
Step 4	Ragi Dosa	2.929	Start with ¼ portion increase gradually	Fry	2–4 min
	Rava Idli	3.120		Steam	15 min
Step 5: No cooking					
Step 5	Yoghurt	7.330	Start with ¼ portion and increase gradually	No Cooking	Through Cheese making
	Srikhand	8.680		No Cooking	
	Cheese	21.28		–	–
	Paneer	21.49			
Step 6: Milk					
Step 6	Pasteurized Milk		Start with ¼ cup and increase gradually		

Legend: The ladder details the recommended portion per serving and cooking time and temperature. G = grams; °C = degree Celsius; min = minutes.

## Data Availability

Data sharing not applicable to this article as no datasets were generated or analyzed during the current study.

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
