# Peer review of "The Need for Culturally Appropriate Food Allergy Management Strategies: The Indian Milk Ladder"

_nutrients, 2023, doi:10.3390/nu15183921_

Round 1

Reviewer 1 Report

This paper adds value for medical practitioners that is relevant to specific geographical and sociocultural context. In the introduction, authors may want to include additional perspective on the milk ladder from other health systems and countries such as the US and other European regions, rather then limiting their comments only on the UK. There is sufficient research and practice application of milk ladder concept in other countries also. The authors should also acknowledge the differing views of researchers and practitioners on the value and practice utility of the milk ladder approach as milk allergy had a natural history of resolution in most children by about 5 years of age. The authors conclusion that the "The Indian Mild Ladder" will be useful tool may be pre-mature given that this has not yet been widely applied in practice and the outcome of such practice application remain unknown for now.

Author Response

Reviewer 1

This paper adds value for medical practitioners that is relevant to specific geographical and sociocultural context. In the introduction, authors may want to include additional perspective on the milk ladder from other health systems and countries such as the US and other European regions, rather than limiting their comments only on the UK. There is sufficient research and practice application of milk ladder concept in other countries also.

Yes, we agree with the reviewer, we write in the 5th paragraph of introduction.

Subsequently many milk ladders were developed based on local dietary pattern and are in clinical use. Recently Mediterranean milk ladder based on the principles of the Mediterranean eating pattern. The protein content of cooked protein in the final food product in each step of the ladder in the matched with that of IMAP ladder. Different recipes for the various steps were provided to increase acceptability and variety. Additionally, the total milk protein content, casein content, and beta-lactoglobulin content of cooked food were estimated by Enzyme-linked immunosorbent assay (ELISA).16 Many modified forms of the milk ladder are widely used internationally and, in some cases, include recipes with local preparation differences.17,18,19 Chomyn, A et al.20 independently proposed Canadian version of cow’s milk and egg food ladders as they felt that European versions of food ladders had limited applicability to the Canadian diet, as they didn’t include foods that are commonly consumed in many Canadian households.

The authors should also acknowledge the differing views of researchers and practitioners on the value and practice utility of the milk ladder approach as milk allergy had a natural history of resolution in most children by about 5 years of age.

We agree hence we write in 4th and 5th paragraph of discussion.

However, we agree that the evidence for introduction of baked milk products in infant’s diet to accelerate development of tolerance is of low quality as highlighted in a systematic review by Labert R et al.37 They cited lack of randomized, controlled trial data and comparator cohort within the study groups of various publications to determine whether or not baked allergen speeds up the resolution of allergy.  They found that studies were mainly observational and merely gave the information that if the child is already tolerant to baked milk, they are more likely to have outgrown their food allergy at any particular time-point than a child who is not tolerant to baked cow’s milk.

Another factor which may hinder clinicians from using milk ladder for reintroduction of baked milk products is the occurrence anaphylaxis. Kim et al.17 reported that 6 % developed mild to moderate anaphylaxis during a challenge but, none of the other studies of baked milk challenges have reported any severe reaction.38,39

The authors conclusion that the "The Indian Mild Ladder" will be useful tool may be pre-mature given that this has not yet been widely applied in practice and the outcome of such practice application remain unknown for now.

Yes, we agree with reviewer that our statement The Indian Milk Ladder will be a useful is little far-fetched as it has not been validated. Hence, we write in the text we hope that the “The Indian Milk Ladder” will be a very valuable resource and helpful tool for the Pediatricians.

Reviewer 2 Report

This is a well written and interesting manuscript based on knowledge of an important clinical problem within food allergy. We know CMPA  affects many children but the authors do not provide any data about prevalence of non-Ige mediated milk allergy especially not from the countries  the study aims to cover. This could be added in the introduction to stress the importance of a culturally adapted milk ladder.

In the aim, the second part is to consider the content of fat and sugar in the foods in the ladder. Foods high in sucrose and fat seems to be common based on the formated group. How were the common foods chosen and why? Are for example foods in step 1 common all over India? Please add more details about this initial development ( even though it is mentioned during 3.1.3)

However, it is not presented in the results how this second aim has been considered and data on content of sucrose and fat is only available in the attachment. I suggest it would be included in the manuscript since it is a part of the aim. Please state content of sucrose if that is what is meant.

The discussion: For other food allergies, the importance of food matrix has been discussed. Since the ladder contains food with different fat content and maybe other factors that may contribute, a section regarding food matrix could be added to broaden the discussion.

Author Response

Reviewer 2

This is a well written and interesting manuscript based on knowledge of an important clinical problem within food allergy. We know CMPA affects many children but the authors do not provide any data about prevalence of non-Ige mediated milk allergy especially not from the countries the study aims to cover. This could be added in the introduction to stress the importance of a culturally adapted milk ladder.

Yes, we agree with the reviewer, we write in the 1st paragraph of introduction

Cow’s milk allergy (CMA) is the most common and complex presentation of food allergy in early childhood. 1-4 Cow’s milk allergy can be divided into IgE and non-IgE mediated forms of food allergy. CMA has an estimated prevalence in developed countries ranging from 0.5% to 3% at age 1 year5. Non-IgE-mediated cow’s milk allergy has a prevalence of less than 1% in children.6 There has been no epidemiologic research on the prevalence of food allergies, including CMA, among Indian children. A hospital-based study from India reported CMA as a cause of malabsorption syndrome in 6% children of all ages and 13% of children below 2 years with chronic diarrhea.7 In another study from India CMA was the cause of chronic diarrhea in 35% of children below 3 years of age.8 Mandal J et al.9 reported milk hypersensitivity in 25% of Indian children aged 5 to 15 years who presented with food allergies.

In the aim, the second part is to consider the content of fat and sugar in the foods in the ladder. Foods high in sucrose and fat seems to be common based on the formatted group. How were the common foods chosen and why? Are for example foods in step 1 common all over India? Please add more details about this initial development (even though it is mentioned during 3.1.3).

Agree, we add more details in materials and methods section

Initial development of the Indian Milk ladder

              The expert group from the different regions in India were asked to provide a list of the foods their patients commonly want to introduce when performing introduction via a ladder approach. The group had a number of teleconferences where the foods were discussed and overlap in food choices were identified in order to develop the final list of foods in the ladder. A list of common foods reflecting the different cultures in India was drawn up, analyzed for their milk protein content, temperature and timing of heating, sugar and fat content (Table 1). The first list of foods included Maida Diamond Biscuits, Gulab Jamun, Ragi/Poha Dosa, Halwa, Rava Idli, Rice/Sago/Semolina pudding, Paneer, Scrambled egg with milk and Rava Laddoo. After discussion of the fat, sugar and allergen content, working (initial) Indian Milk Ladder was designed by consensus (Table 1). Foods which are prevalent across different geographical locations and cultures were included. 

However, it is not presented in the results how this second aim has been considered and data on content of sucrose and fat is only available in the attachment. I suggest it would be included in the manuscript since it is a part of the aim. Please state content of sucrose if that is what is meant.

Agree with the reviewer, we write as follows

3.1.3 Adaptation of sugar content

              The foods identified by the expert group were typically high in fat and sugar. SHR and ED spent considerable time to reduce the sugar and fat content of foods to be more acceptable according to nutrition standards but still palatable.  The DG USA recommends less than 10% kcal from sugar for children over 2 years which is 25 g sugar per day. We have reduced the sugar content as indicated in table 1.

Table 1: Adapted sugar Content of foods high in sugar

Milk ladder foods

Sugar per portion in original recipes

Sugar content in adapted recipes

Reduction in sugar per portion

Maida diamond biscuit

1.5 g

0.75 g

0.75 g

Milk cookie

11.25 g

1.9 g

9.6 g

Gajar halwa

25 g

6 g

19 g

Gulab Jamun

20 g

5 g

15 g

Rasagulla

33 g

13 g

20 g

Ragi Sair/Kanji

5 g

5g

Nil

Rice Kheer

31.25 g

12.5 g

18.75 g

Srikhand

30 g

7.5g

22.5 g

The discussion: For other food allergies, the importance of food matrix has been discussed. Since the ladder contains food with different fat content and maybe other factors that may contribute, a section regarding food matrix could be added to broaden the discussion.

This has been included in the following section.

3.1.4 Consideration of food matrix

Bloom et al.28 showed that a wheat matrix, decrease IgE antibody binding to milk proteins. In addition, chemical reactions between proteins and fat and sugars in recipes may also reduce allergenicity.29 Bacteria used in yogurt production may also affect allergenicity but has not been well studied. Our awareness of all these factors led to not only calculating the milk content of the ladder but also scientifically quantifying the milk content of the ladder foods within their complex matrices.